# Polyoxometalate–Polymer Hybrid Materials as Proton Exchange Membranes for Fuel Cell Applications

**DOI:** 10.3390/molecules24193425

**Published:** 2019-09-20

**Authors:** Liang Zhai, Haolong Li

**Affiliations:** State Key Laboratory of Supramolecular Structure and Materials, College of Chemistry, Jilin University, Changchun 130012, China; dijing18@mails.jlu.edu.cn

**Keywords:** polyoxometalates, hybrid materials, polymer electrolytes, proton exchange membranes, proton transport, fuel cells

## Abstract

As one of the most efficient pathways to provide clean energy, fuel cells have attracted great attention in both academic and industrial communities. Proton exchange membranes (PEMs) or proton-conducting electrolytes are the key components in fuel cell devices, which require the characteristics of high proton conductivity as well as high mechanical, chemical and thermal stabilities. Organic–inorganic hybrid PEMs can provide a fantastic platform to combine both advantages of two components to meet these demands. Due to their extremely high proton conductivity, good thermal stability and chemical adjustability, polyoxometalates (POMs) are regarded as promising building blocks for hybrid PEMs. In this review, we summarize a number of research works on the progress of POM–polymer hybrid materials and related applications in PEMs. Firstly, a brief background of POMs and their proton-conducting properties are introduced; then, the hybridization strategies of POMs with polymer moieties are discussed from the aspects of both noncovalent and covalent concepts; and finally, we focus on the performance of these hybrid materials in PEMs, especially the advances in the last five years. This review will provide a better understanding of the challenges and perspectives of POM–polymer hybrid PEMs for future fuel cell applications.

## 1. Introduction

Burning fossil fuels is the principal energy-producing pathway of the world, which emits a large amount of greenhouse gas like carbon dioxide and also environmental pollution from sulfur and nitrogen oxides. The aggravation of environmental problems and the energy crisis have accelerated much research on renewable energy technologies. Fuel cells are electrochemical energy conversion devices that can convert chemical energy directly into electrical energy through a pair of redox reactions, offering extraordinary superiorities in fuel sources, power density, energy conversion efficiency and environmental compatibility. Thus, they have received worldwide attention for their wide potential in transportation and in stationary and portable power applications [1,2]. Among various types of fuel cells, proton exchange membrane fuel cells (PEMFCs) have attracted intensive attention in the past decades. The proton exchange membrane (PEM), serving as a conductor for protons and a barrier for fuels, oxidants and electrons plays the central role in determining the performance of them. Thanks to their easy processing, excellent toughness and durability, solid-state polymer PEMs are mostly applied for fuel cells. These membranes should meet the following requirements: high proton conductivity; low fuel permeability; excellent mechanical, chemical, electrochemical and thermal stabilities; sufficient long-term durability; easy fabrication for membrane electrode assembly and a competitive low-cost [3]. 

Currently, the commercially available perfluorosulfonic acid ionomer membranes (PFSA), represented by Nafion produced by Dupont Co., are predominantly used in these fuel cells and have served as the benchmark for membrane performance due to their excellent chemical stability as well as high proton conductivity under wet conditions. However, disadvantages like the high cost, high methanol permeability, low operating temperature and low proton conductivity at low humidity restrict their further use [4,5]. For decades, much effort has been expended in developing novel membranes with lower production costs and improved attributes, such as modification of Nafion, synthesis of sulfonated hydrocarbon membranes and the formation of organic–inorganic hybrid membranes. In particular, organic–inorganic hybrid membranes provide a unique combination of both the properties of two components which are highly expected to satisfy the aforementioned demands and overcome some of the drawbacks of Nafion membranes [6,7,8]. 

Polyoxometalates (POMs), the rigid, compact, nanosized inorganic building blocks can be reliably utilized in the formation of functional hybrid and nanocomposite materials [9,10]. One of the main groups of POMs, heteropolyacids (HPAs), displaying strong Brönsted acidity and the highest proton conductivity in their fully hydrated state among inorganic solids near ambient temperatures, are promising candidates for fabricating composite PEMs. A large number of bonded water molecules, high thermal stability, structural flexibility and mobility of HPAs are beneficial for fuel cell applications as well [11,12,13,14]. Nevertheless, the high water solubility of HPAs leading to the progressive leakage from membranes and the decrease of proton conductivity is a well-recognized obstacle to using HPA-based PEMs. 

Here, we summarize a number of research works on the progress of POM–polymer hybrid materials, especially POM–polymer PEMs. Special attention is given to the proton-conducting properties of HPAs and the fabrication strategies of class-I and class-II POM–polymer hybrid materials according to the interactions between POMs and polymer matrices are briefly reviewed. The applications of these strategies in preparing POM-based PEMs and the approaches to reduce HPA leakage are elaborated in the following section. The challenges remaining and future directions of POM–polymer PEMs are also discussed.

## 2. Advantages of Polyoxometalates (POMs) in Proton Exchange Membranes (PEMs)

POMs are a large class of discrete, molecularly defined early transition metal oxide clusters [9]. Owing to the great diversity of nuclearities, sizes and structures, they exhibit unique properties, such as strong acidity, oxygen-riched surfaces, electron-accepting capability, chemical adjustability and so on, being promising building blocks in preparing functional materials [10,15]. Given a number of reviews have been published about the applications of POMs and POM-based materials in various areas [9,10], in this part we focus more on the use of POMs as solid proton conductors, especially the proton-conducting properties of HPAs. 

POMs, formed by the connection of {MO_x_} polyhedras (M = addenda atom, usually tungsten, molybdenum or vanadium in high oxidation state) can be divided in two generic families: heteropolyoxometalates (Figure 1) and isopolyoxometalates according to the existence of heteroatoms [16]. Due to the delocalization of the negative charge over many atoms of the polyanion and the double-bond character of M=O_d_ inducing less charge distribution over the outer surface, HPAs show very strong Brönsted acidity and POMs are suitable as proton carrier media [17]. Nakamura et al. first reported the extremely high proton conductivity of 0.18 S cm^−1^ of H_3_PMo_12_O_40_·29H_2_O and 0.17 S cm^−1^ of H_3_PW_12_O_40_·29H_2_O in 1979 [11] and afterwards the proton-conducting properties of heteropoly compounds, A_3_PM_12_O_40_·nH_2_O, where A = H, Li, Na, K, Rb, Cs or NH_4_ and M = Mo or W were studied by Korosteleva et al. [18]. These pioneer works opened up a new field of POM as a proton conductor. Among various POMs, the most widely available and studied are of Keggin type, especially phosphotungstic acid (PWA). 

There are two categories of protons existing in HPAs. One is the dissociated, hydrated proton combined with the heteropolyanion, the other is the unhydrated acidic proton associated with the structural oxygen atoms [19,20]. In the anhydrous state, the positions of the acidic protons of PWA were determined to be on both O_c_ and O_d_ of the Keggin unit by Janik and Deng et al. theoretically and experimentally [20,21]. The elementary step for proton motion in anhydrous PWA is hopping between surface oxygen atoms in the Keggin unit or between neighboring Keggin anions [22,23,24]. A DFT computational study by Janik et al. showed the possible mechanisms of proton mobility based on structural rearrangements and the calculated activation barrier for proton hopping between neighboring surface oxygens in the Keggin anion of anhydrous PWA [23]. Kolokolov et al. then clarified the details of proton mobility in anhydrous PWA over a wide temperature range experimentally by the first time [24]. They found that two types of surface protons existed, the ones localized on bridged oxygen sites, flipping rapidly, and the ones migrating relatively fast over the surface oxygens of the Keggin unit. Few of these protons kept that fast migration property below 150 °C, but almost all of them diffused rapidly over the surface by hopping between neighboring oxygens when temperature was over 230 °C.

HPAs usually crystallize with a large number of water molecules and form the secondary structure, resulting in the formation of new active species such as the [H_3_O]^+^ and the [H_5_O_2_]^+^ ions (Figure 2) [12,23]. The heteropolyanions are quite mobile in HPA crystals. Due to such structural flexibility, not only water but also various polar molecules can enter and leave HPA crystals and HPAs exhibit an extremely high proton mobility and a “pseudoliquid phase” [25]. Thus, the degree of hydration which is very sensitive to temperature and relative humidity plays a significant role in structural changes, protonic-species (OH^-^, H_2_O, H_3_O^+^, H_5_O_2_^+^) equilibrium, proton mobility and the mechanisms of proton transport and is further reflected in proton conductivity [22,26,27,28].

In the hydrated state, it can be divided into three types of water: hydration water (the crystalline water combined without protons), protonized water (the crystalline water combined with protons), and the structural water formed by protons and oxygens during decomposition of HPAs [29]. According to the systematic studies of Mioč and co-workers, during the dehydration process, H_3_PW_12_O_40_·29H_2_O (triclinic) underwent two symmetry transformations to H_3_PW_12_O_40_·21H_2_O (orthorhombic) at around 30 °C and then to H_3_W_12_PO_40_·14H_2_O (triclinic) at 40 °C. After the fast phase transition of H_3_PW_12_O_40_·14H_2_O (triclinic) to H_3_PW_12_O_40_·6H_2_O (cubic) at about 60 °C, the hexahydrate remained stable up to 170 °C. Then H_3_PW_12_O_40_·6H_2_O lost 5.8–6 molecules of water in two steps between 170 and 240 °C and became anhydrous phase, H_3_PW_12_O_40_, in which no more crystalline water existed, but only free protons. The anhydrous phase was stable up to 400 °C. In the range of 410–440 °C, a half or one water molecule, formed by acidic protons and oxygens from the host lattice lost, which resulted in the formation of denuded Keggin anion. Moreover, this process was more or less common to all isostructral heteropoly compounds of Keggin-type (12-molybdophosphoric, 12-tungstosilicicacid) they investigated: the dehydration process was finished about 250 °C and the hexahydrates were the more stable phase [28,30,31]. It should be noted that despite the Keggin unit being preserved, in accordance with the results of Fourier transform infrared (FT–IR) spectroscopy and microcalorimetry on water sorption, changes in hydration state were reversible with the exposure to gas-phase water only between 100 °C and 200 °C [20]. The excellent thermal stability and water-retention capability make HPAs potential for using in high-temperature and low-humidity PEMFCs [14].

For low hydration levels (n < 10), the Keggin units form a cubic lattice with water molecules, which is represented by H_3_PW_12_O_40_·6H_2_O, the most stable hydrate at normal conditions and the only hydrate whose structure (including the proton positions) was resolved by neutron diffraction experiments, in which the polyanions at the lattice points of a body-centered cubic (bcc) structure are connected by H_5_O_2_^+^ (formed by the water molecules and the acidic protons) along the cubic faces [20,32]. Subsequent studies using inelastic neutron scattering, infrared spectroscopy and Raman spectroscopy showed that both [H_3_O]^+^ and [H_5_O_2_]^+^ ions existed in the hexahydrate and it offered a possible temperature-dependent dynamic equilibrium of the H_5_O_2_^+^, H_3_O^+^,H_2_O and H^+^ [33,34]. Kolokolov and co-workers reported a detailed ^2^H nuclear magnetic resonance (NMR) study on the proton and water mobility in different hydration level of HPA hydrates [35]. The fast internal rotations about their symmetry axes of H_2_O and H_3_O^+^ were not affected by the water concentration in the case of n < 10. In addition, H_3_O^+^ might involve in the process of the decomposition of itself back to water molecules and the surface acid protons and moved wobblingly on the surface of the Keggin unit, which might determine the proton diffusion and the charge transfer mechanism. Because the formation of less mobile [H_5_O_2_^+^] ions was favored in high-level hydrates, the rate constant of the migration between neighboring Keggin units quite relied on the water concentration, which was consistent with the research of Slade et al. [36]. DFT studies of the motion of protons concluded that the adsorption of water could enhance the proton mobility and substantially reduce the energy barrier of proton hopping by an order of magnitude and with low amounts of water, proton transport was dominated by the vehicle mechanism as expected [23]. For higher hydration levels (n > 10), it is generally accepted that the polyanions are connected by a hydrogen-bond network formed by water molecules [35]. In this case, the most likely proton transfer mechanism is of the Grotthuss type proceeding by any necessary reorientations, e.g., the 180° flips of water, the uniaxial rotation of the H_3_O^+^ ions and the isotropic motion of H^+^ in H_3_PW_12_O_40_·14H_2_O [33,36,37,38]. 

The remarkably high proton conductivity of 0.18 S cm^−1^ of H_3_Mo_12_PO_40_·29H_2_O and 0.17 S cm^−1^ of H_3_PW_12_O_40_·29H_2_O (pellet samples) at 25 °C were measured by a graphite cell [11]. Later, Hardwick et al. using alternating current (AC) conductivity and pulsed ^1^H NMR investigated the 21-hydrates of H_3_PW_12_O_40_ and H_3_Mo_12_PO_40_ [39]. Slade et al. also using these two techniques studied the temperature-dependent proton conductivity and ^1^H self-diffusion coefficients of H_3_PW_12_O_40_·nH_2_O (n = 6, 14, 21) [36]. Both the temperature-dependent AC conductivity and the self-diffusion coefficients showed an Arrhenius form in the temperature range. They explained that the differences of proton conductivity between different measurements might relate to the mechanism of ^1^H self-diffusion which had much stronger temperature dependence at lower temperatures (T < 7 °C). They further used a combination of AC and direct current (DC) techniques to characterize the proton conductivity in pelletized hexahydrates of H_3_PW_12_O_40_ and H_4_SiW_12_O_40_ [40] and the proton conductivity of H_3_PW_12_O_40_·6H_2_O at elevated temperatures [41]. Also, dielectric relaxation spectroscopy has been used for the bulk proton conductivity of pelletized H_3_PW_12_O_40_·6H_2_O [42]. The studies mentioned above were conducted on pressed powders of HPAs, which exhibited high water partial pressures and short-circuiting of the bulk conductivity by fast conduction paths. So Kreuer et al. [43] and Padiyan et al. [44] prepared single crystals of defined hydration levels to study their proton conductivity. Due to different procedures of sample preparing and measurements, the values of proton conductivity at the same hydration level may have some differences, but they were found to decrease with the decreasing hydration levels. Details are listed below (Table 1).

Besides the remarkably high proton conductivity, excellent thermal stability and water- retention capability, POMs, the rigid inorganic clusters with electrostatic cross-linking ability, can also perform as nanoenhancers to enhance the modulus of polymers [45,46,47]. The contrast in electron density between POMs and polymers makes the observation of microscopic structures direct and convenient without any additional staining, which greatly contributes to the study of the self-assembling process in POM–polymer composite materials [48]. Furthermore, the fast reversible multielectron redox transformations and tunable redox properties of POMs endow them with stability during electrochemical use [49]. All of these advantages make POMs promising components in preparing PEMs. It worth noting that types of POMs introduced to PEMs are not limited to typical Keggin-type POMs like PWA, Dawson-type [19,50,51] and some vanadium-substituted Keggin-type POMs [52,53,54] are gradually developed. 

## 3. Fabrication Strategies of POM–Polymer Hybrid Materials

Organic–inorganic hybrid materials, integrating both the properties of two components have attracted extensive attention in the development of functional materials. Thanks to their easy processing, excellent toughness and durability, polymers are usually suitable matrices to introduce POMs and exhibit synergistic functionalities [55]. Conventional organic–inorganic hybrids can be divided into two categories according to the interactions between organic and inorganic moieties [15]. Class-I hybrids can be assembled through non-covalent interactions, such as electrostatic interactions, hydrogen bonds and van der Waals interactions. Strong covalent bonds lead to the formation of class-II hybrids. Note that covalently modified POMs still have the capability for further integration via non-covalent interactions [10]. Because some researchers in this area have already summarized the progress of both strategies [10,15,48,55,56,57,58], herein we select some representative or latest research works to simply illustrate the fabrication strategies of both classes of POM–polymer hybrid materials. The applications of these strategies in preparing PEMs will be demonstrated in detail in Section 4.

### 3.1. Class-I POM–Polymer Hybrids 

In general, the Class-I POM–polymer hybrids are based on electrostatic interactions and hydrogen bonding between POMs and polymer matrices. For water-soluble polymer matrices such as poly(vinyl alcohol) (PVA), poly(ethylene glycol) (PEG) and poly(vinyl pyrrolidone) (PVP), solution blending is preferable to fabricate their hybrid materials with POMs. When using strong acidic POMs, there are usually hydrogen bonds or electrostatic interactions existing between POMs and polymer matrices that can stabilize the resultant hybrids. For example, Uchida and Yin et al. dissolved H_3_PW_12_O_40_·nH_2_O, PEG1000 and CsNO_3_ into water and obtained highly efficient proton-conducting POM–PEG crystalline hybrid materials, in which a single PEG chain was confined in the one-dimentional nanochannels defined by the frameworks of POMs [59,60]. Besides water, polar organic solvents such as alcohols and N, N-dimethylformamide which are able to make POMs and polymers a homogeneous solution, are also widely used in solution blending [61,62,63]. It should be noted that when dealing with water-insoluble polymers which cannot directly blend with POMs, solution swelling is also a constructive way to achieve the hybridization process. For instance, Jiang et al. immersed mesoporous Nafion membranes in PWA solutions then the Nafion membrane impregnated with PWA was employed as a central layer sandwiched with two side block layers to obtain a multilayer PEM [64]. The resultant PEM with the PWA loading of 19 wt% possessed a much higher conductivity of 0.072 S cm^−1^ at 80 °C under 40% relative humidity (RH). 

Although solution blending is frequently used in fabrication of class-I POM–polymer hybrids for its convenience and wide range of application, POMs are prone to agglomerate to form large particles during the blending process which may cause phase separation between POMs and polymer matrices and lower performance of the composites. In order to achieve more precise assembly, POMs as a type of anionic cluster can be assembled with cationic polyelectrolytes in which POMs are deposited as molecular-level monolayers by the layer-by-layer (LBL) self-assembly techniques (Figure 3) [65,66]. LBL self-assembly rests on the alternating deposition of oppositely charged substances on a substrate surface and the formation of the thin multilayer film is primarily attributed to non-covalent interactions [67]. Li et al. deposited different numbers of PAH/GO/PAH/PWA (where PAH is poly(allylamine hydrochloride), GO is graphene oxide) layers on modified substrates to construct GO–PWA multilayer films [68]. Photocatalytic activity of PWA induced GO to convert to reduced GO (rGO) under ultraviolet (UV) irradiation. The field effect transistors based on the composite films exhibited good transport properties for both holes and electrons.

Due to the high crystal energies of the POM structure, POMs show poor compatibility with most polymers. Surface modification by the exchange of countercations on the POMs for cationic surfactants can dramatically improve the surface chemical properties of POMs and make the surfactant-encapsulated POM clusters compatible with the organic moieties [69]. The surfactant can be cationic terminated polymer chains [70], as well as cationic monomers [71] or cationic chain transfer agents then could be used for post-polymerization [72]. Recently, our group synthesized a set of POM-based mobile-ligand nanofillers by reversible addition-fragmentation chain-transfer (RAFT) polymerization [47]. These POM-based nanofillers were composed of a [SiW_12_O_40_]^4−^ core and four polystyrene chains with increasing length, which exhibited intriguing multiscale hierarchical self-assembly behavior both in their bulk state and in additional polymer matrices.

### 3.2. Class-II POM–Polymer Hybrids

Lindqvist, Keggin, Anderson and Dawson, these four types of POMs are usually used in the fabrication of class-II POM–polymer hybrids. Normally, the reactions of covalently modified POMs involve silanization, imidization and esterification [58]. 

Research on the synthesis and characterization of organically functionalized POMs anchored to a polymeric backbone was first reported by Judeinstein in 1992 (Figure 4) [73]. Four trichlorosilane-modified lacunar [SiW_11_O_39_]^4−^ precursors were synthesized. After free radical polymerization of these difunctionalized POM-based monomers, POM-based polymers with linear or branched structures were obtained. Later, this approach was extended by Mayer and Thouvenot et al. to the divacant heteropolyanion [γ-SiW_10_O_36_]^8−^ [74,75]. Moreover, Herring group devoted to developing class-II PEMs in which the silanization-modified mono-lacunary 11-silicotungstic acid was the only proton-conducting component [76,77]. These covalently immobilized proton conductive membranes effectively avoid the leakage of HPAs.

Lindqvist-type POMs, [Mo_6_O_19_]^2−^ can be modified by the substitution of Mo=O bonds for Mo≡NR bonds through imidization (Figure 5). The styrylinido ligand in the Lindqvist derivative [Mo_6_O_18_(NC_6_H_4_CH=CH_2_)]^2-^ were successfully synthesized by Maatta et al. [78]. However, hybrid polymers with covalent attachment of POMs have been limited to insulating non-conjugated polymers until the works of Peng group. They developed a more simple and efficient method for synthesizing imido derivatives of [Mo_6_O_19_]^2−^ by the addition of dicyclohexylcarbodiimide [79]. On this basis, the first two sets of hybrid conjugated polymers with poly(phenylene ethynylene) as backbone and POM clusters as side-chain pendants were synthesized and their electronic, optical and electrochemical properties were studied [80]. Additionally, bifunctionalized imido derivatives of hexamolybdate with two iodo or ethynyl groups and theirs containing main-chain polymers were successfully synthesized [81,82]. 

Esterification between tris(hydroxylmethyl)-type derivatives and the terminal oxygen atoms of Dawson-type POM ([P_2_V_3_W_15_O_62_]^9−^) or Anderson-type heteropolymolybdate is another widely used method for covalent modification (Figure 6). A monomer synthesized by reacting tris(hydroxymethyl)aminomethane (Tris)-modified [P_2_V_3_W_15_O_62_]^9-^ with methacryloyl chloride was copolymerized with methyl methacrylate and the properties of the obtained POM–polymer hybrids as photoresists were tested [83]. The applications of controllable polymerization to the Tris-type [P_2_V_3_W_15_O_62_]^9−^ modified norbornene monomer was reported by Wang group [84]. The polymers obtained yielded a well-defined linear side-chain structure with relatively high molecular weights and low polydispersities and showed a good catalytic performance. The symmetrical structure of Anderson-type POMs endows them with convenience in organic modification and is beneficial for the formation of main-chain POM-containing polymers (Figure 6b). Mn-Anderson clusters with a coumarin-Tris on each side which could undergo reversible light-driven polymerization to obtain cyclobutane dimers was designed by Song et al. [85]

Apart from the three methods mentioned above, organotins and post-modification are also used in the preparation of class-II POM–polymer hybrids. Methods classified by different organic groups covalently linked to POM units were discussed in detail in the review of Dolbecq et al. [15]. In terms of the topology of the resultant polymers containing organically modified POMs, it can be divided into different kinds such as side-chain, main-chain, and head-tail, in which representative works were summarized by Wang et al. [48] and Yan et al. [58]. 

Although covalent modification can provide precise anchoring sites and avoid the leakage of POM components, the complicated synthesis processes and required non-conventional purification techniques make it quite challenging to fabricate class-II POM–polymer composites compared to those non-covalent POM–polymer hybrids. Furthermore, limited variation of POMs and modified chemistry slow down the development of class-II POM hybrids [58]. So the majority of research works on the POM-based PEMs prefer to choose the non-covalent method.

## 4. POM–Polymer Hybrid PEMs

### 4.1. Class-I POM–Polymer PEMs

Polymers with charged or polar groups are usually hybridized with POMs by various approaches through electrostatic interactions or hydrogen bonds to develop class-I PEMs. These polymer matrices mainly involve commercially available Nafion membranes, polybenzimidazole, chitosan, hydrocarbon membranes like polyethersulfone, poly(ether ether keton), poly(arylene ether ketone), or their sulfonated forms, and water-soluble polymers like PVA, PEG and PVP. POMs can be directly doped into these matrices or with other inorganic moieties together [86,87,88]. 

However, PEMs fabricated by direct blending those polymers with HPAs result in inevitable leakage caused by weaker binding forces between HPAs and polymer matrices and the high water solubility of HPAs. The strategies having been applied to stabilize HPAs in class-I POM–polymer PEMs can be summarized in two categories. One is using water-insoluble salts of HPAs by partially exchanging the protons with large cations like Cs^+^, NH_4_^+^, Rb^+^ [89,90,91,92]. The other is immobilization of HPAs onto various substrates including SiO_2_ [93,94,95], ZrO_2_ [96], mesoporous silica [97,98], carbon nanotubes [99,100], glass fiber [101,102] and GO or rGO [52,53,54,103,104]. The porous structure, hygroscopicity, the exceptionally high specific surface area and the surface polar groups (–O–, –OH, –COOH) of those substrates can further stabilize HPAs and enhance the thermal stability and mechanical properties as well. In order to achieve better distribution of HPAs and control for the size of the particles when using SiO_2_ or mesoporous silica as substrates, an in situ sol-gel reaction is normally utilized [93,97,105,106]. Notably, amino-containing polymers can be grafted onto substrates [107], blended with other polymers mentioned above [61,108] or used alone [109,110] to stabilize HPAs by hydrogen bonding, electrostatic force, or acid–base interactions. For example, Xiang and Lu et al. developed a self-anchored HPA hybrid PEM by fixing PWA into PVP and polyethersulfone matrix, which exhibited a high proton conductivity of 66 mS cm^−1^ at 60 °C with 100% RH and a good stability for a test period of more than 500 h [61]. Using a triblock copolymer P123 as a soft template, they prepared a chitosan membrane with a submicro-porous structure and further fabricated the chitosan–PWA hybrid PEM [110]. The sole proton conductor, PWA was confined in the submicro-porous structure cross-linking the polymer matrix, which not only enhanced proton conductivity and mechanical properties but also reduced methanol permeability and the leakage of itself. Recently, Guiver and co-workers reported a magnetic field induced composite PEM containing through-plane-aligned proton channels with PWA arrays inside (Figure 7a) [62]. The PEM was co-casted by polysulfone and a water-insoluble proton-conducting paramagnetic complex formed by both water-soluble ferrocyanide-coordinated poly(4-vinylpyridine) (CP4VP) and PWA via electron transfer under the magnetic field. At the same time, the proton-conducting paramagnetic complex oriented in the through-plane direction, creating short-pathway highly proton-conducting channels. Particularly, the magnetic field obviously improved the dispersion of PWA and no agglomeration appeared in the induced membrane (Figure 7b). Moreover, ferrocyanide groups tethered to the polymer, via redox cycles could continuously consume electrophilic OH· and OOH· radicals during use. These designs overcame both the leakage of PWA and radical attack, endowing the composite membrane outstanding proton conductivity with a value of 215 mS cm^−1^ at 95 °C in water, excellent cell performance and strong stability for a period of 30 days without degradation. The field-inducing conduction pathway built in this work opens up a new strategy to construct stable POM-based nanostructured PEMs and stresses the importance of less tortuous proton conducting pathway to PEMFC performance.

The self-assembly of block copolymers into well-ordered nanostructures providing tunable morphologies and domain sizes is contributed to investigating the relationship between the morphology and transport properties [111]. Generally, block copolymers for PEMs contain hydrophilic blocks facilitating proton transport and hydrophobic blocks providing mechanical support. For HPA-containing sulfonated block copolymers [104,112,113,114], HPAs can serve as hygroscopic fillers and the extra proton conductor to boost the conductivity at low-humidity and high-temperature conditions. Also, HPAs can generate hydrogen-bond networks with the polar groups (–O–, O=S=O, –SO_3_H, –OH) of the polymer backbone which enable the proton conduction to be faster. Similar to those block copolymers, graft copolymers with sulfonated side chains attaching to the hydrophobic main chain which structurally mimic PFSA can also embed HPAs to fabricate organic–inorganic hybrid PEMs [115]. Aside from the intrinsic high proton conductivity and the properties as inorganic nanofillers, HPA as a macroion can control the morphology of block copolymers by electrostatic-crosslinking [46]. Li et al. utilizing H_4_SiW_12_O_40_ (SiW) induced the phase transition of poly(styrene-block-2-vinylpyridine) (PS-b-P2VP) from an initial lamellar phase to a stable bicontinuous phase (Figure 8). Due to the formation of continuous conductive domains by the incorporation of excellent proton conductors, the proton conductivity of the bicontinuous polymer nanocomposite was greater than the original lamellar one by two orders of magnitude. Although the proton conductivity of the nanocomposite in this work is far from the standard of PEMs, this concept of using POMs as conductive additives to obtain a stable bicontinuous phase can be extended to other polymer systems. Also, the continuous conductive domains can be regarded as network-like proton conducting pathways, which will help further understand the impact of nanostructures on transport properties.

By contrast with the above direct blending of POM-based PEMs, the electrostatic self-assembly LBL technique can efficiently control the microstructure of composite membranes, avoid the aggregation of additives and improve the uniformity of them [66]. For, example, Yang et al. self-assembled PWA and poly(diallyldimethylammonium chloride) onto the Nafion membrane surface [116]. Similarly, Na group constructed stable multilayers by self-assembling polycation such as chitosan [117], polypyrrole (PPY) [118] and polyaniline (PANI) [119] with PWA on the support materials, sulfonated poly(aryl ether ketones)-bearing carboxylic pendant groups (SPAEK-C). The PWA–chitosan and PWA–PPY multilayers helped to restrict the hydrophilicity of the SPAEK-C and block the methanol transport pathways thus suppressed the water swelling and methanol permeation. Because PANI and PWA had strong affinity to water which enlarged the hydrophilic regions, PANI–PWA enabled easier proton transfer and higher proton conductivity but little impact on water uptake [119]. The proton conductivity of PEMs prepared by the LBL self-assembly method usually decrease with the increase in the number of bilayers for the restriction of charge carrier species. The addition of proton-conducting PWA in the multilayer structure and optimized nature of the polyelectrolyte bilayers will minimize the reduction in proton conductivity. Additionally, with the strong electrostatic force, PWA can be trapped in the PWA-containing bilayers so that it reduces the leakage to obtain stable PEMs and improve the cell performance. 

Besides aforementioned frequently-used methods for class-I POM–polymer PEMs, polymerization of encapsulating HPA with polymerizable ionic liquid [120] or introducing HPA-loaded polymer microcapsules to a polymer matrix [121] were also reported.

### 4.2. Class-II POM–Polymer PEMs

The other effective solution to avoid leakage of HPA is covalent attachment to polymer backbones [76,77,122,123]. Herring group have devoted to the development of PEMs based on lacunary Keggin type HPA, divinylsilyl-11-silicotungstic acid. This HPA was copolymerized with butyl acrylate (BA) and 1,6-hexanediol diacrylate (HDDA) (Figure 9) [76]. The polymer obtained was then casted into films under UV light. Results of the structure and morphology indicated that the materials were amorphous but phase-separated with HPA clustering in the volume of 75%, which was the first reported HPA-containing free-stand membranes with such a high loading of HPAs. Due to robust covalent bonds, only small organic oligomers were lost in liquid water and the degree of swelling was obviously lower than a typical PFSA. DC measurement showed high proton conductivity of 170 mS cm^−1^ at 80 °C and outperformed Nafion 112 at all temperatures with 100% RH. Fast proton transport in this hybrid material was facilitated by as few as three water molecules per acid so that the membrane could contain highly mobile protons in hotter and drier conditions [77].

Recently, through a three-step synthesis, they covalently immobilized 11-silicotungstic acid on a commercial fluoroelastomer (the final product called PolyHPA) in which HPA served as both the radical decomposition catalyst and the proton conductor (Figure 10) [123]. By contrast with previous approaches to improve the chemical stability of PFSA by treating it with elemental fluorine to minimize the number of reactive carboxylic acid end groups or adding radical decomposition catalysts such as CeO_2_ and MnO_2_, the covalently attached HPAs as radical decomposition catalysts can achieve a trade-off between durability and performance. Despite HPAs clustering, which was hypothesized to reduce the proton transport under low humidity because of the non-continuous morphology, this material showed very high proton conductivity of over 200 mS cm^−1^ at 80 °C, 95% RH. It also showed superior chemical stability either in the aggressive chemical accelerated stress test or fuel cell testing. 

Compared to non-covalent blending PEMs which are limited to the leakage of HPAs and the stability of membranes, the class-II POM–polymer PEMs can overcome the challenges of chemical degradation and leakage, simultaneously, and achieve a higher concentration of HPAs, thus reducing the isolation of HPAs and allowing close enough proximity to each other to easily exchange protons through the membrane [122]. However, the high concentration of HPA may make the morphology of those membranes unclear and affect research on the relationship between the morphology and proton transport.

## 5. Conclusions and Perspectives

POMs containing protons as the countercations have been extensively used as multifunctional proton conductors. Except for their intrinsic high proton conductivity, POMs can also serve as electrostatic crosslinkers, nanoenhancers and radical decomposition catalysts etc. These features are in high demand for PEMs to achieve good conductivity as well as enhanced mechanical and chemical stabilities, thus enabling POMs to be suitable inorganic building blocks in the fabrication of POM–polymer hybrid PEMs. In past decades, POMs have been combined with polymer moieties through non-covalent and covalent interactions to prepare various hybrid materials for PEMs, where the distributed morphology of POMs can be precisely controlled down to a several nanometers scale; meanwhile, the proton transport efficiency and the leakage issue of POMs are gradually improved.

Despite considerable progress, there are still many challenges and opportunities for continued advances in POM–polymer hybrid PEMs. From the aspect of structural design, a rational optimization of the chain topology of polymer moieties through mature linking chemistry is crucial to control the microphase-separation of charged and neutral domains of polymer moieties, which can further influence the morphology of proton transport nanochannels formed by POM–polymer hybrid domains, thus determining the final proton conductivity. From the aspect of proton transport mechanisms, the relationship between the packing structure of POMs in the polymer domains or nanochannels and their proton transport ability is worth exploring in depth, especially the confinement effect of nanochannels on the proton transport process. From the aspect of enhanced PEM functions, the performance of POM–polymer hybrid PEMs under high temperature and low humidity and even the anhydrous condition requires further improvement. Moreover, the synergistic effect of additional POM properties like radical decomposition catalysts to enhance the stability of PEMs during operation needs to be investigated comprehensively. As a whole, it can be envisioned that POM–polymer hybrid materials will be developed as a promising PEM candidate for future fuel cell applications. 

## Figures and Tables

**Figure 1 molecules-24-03425-f001:**
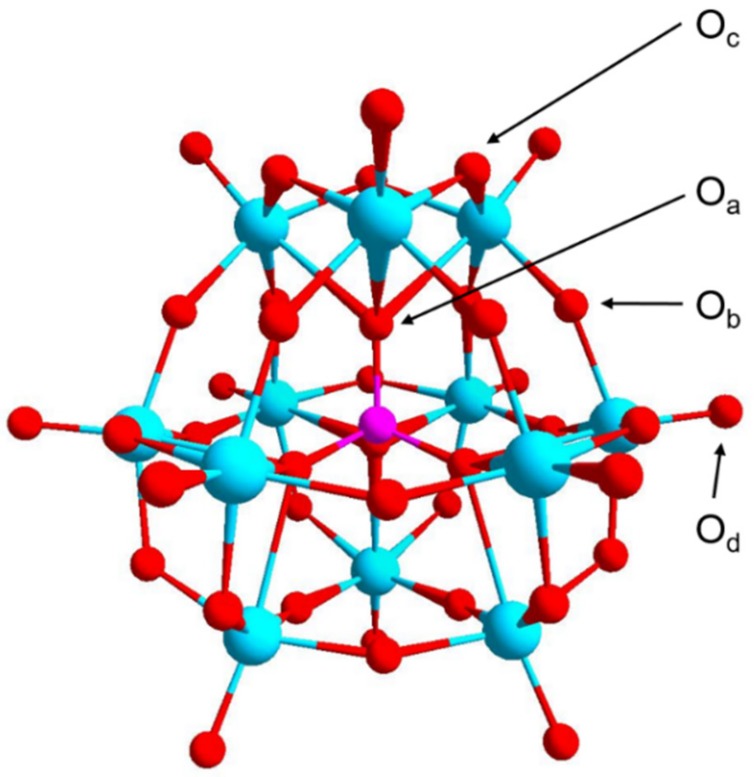
The structure of Keggin-type polyoxometalates (POMs), taking PW_12_O_40_^3−^ as an example, identifying the four types of oxygen in the structure: the central oxygen atom (O_a_); the bridging oxygen atom that bridges two tungsten atoms not sharing a central oxygen atom (corner-sharing, O_b_); the bridging oxygen atom that bridges two tungsten atoms sharing a central oxygen atom (edge-sharing, O_c_) and the terminal oxygen atom (O_d_). P, pink ball; W, blue ball; O, red ball. Reproduced with permission from [20]. Copyright (2003) Elsevier.

**Figure 2 molecules-24-03425-f002:**
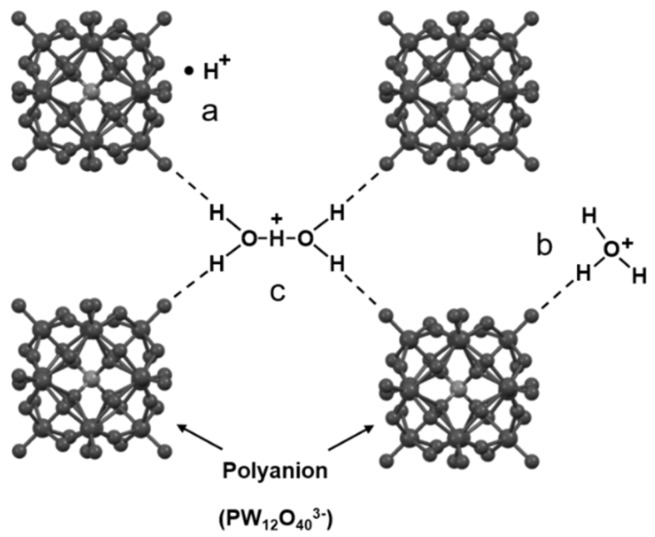
Protonic species in H_3_PW_12_O_40_·nH_2_O (0 < n < 6); (**a**) the isolated acidic proton; (**b**) H_3_O^+^ and (**c**) H_5_O_2_^+^. When 0< n < 6, acidic protons are located randomly (may be in H_5_O_2_^+^ bridges, H_3_O^+^ or bonded to the oxygens of the Keggin unit); when n = 6, the acidic protons are located in the central H_5_O_2_^+^ bridges between lattice points. Reproduced with permission from [12]. Copyright (2011) Springer.

**Figure 3 molecules-24-03425-f003:**
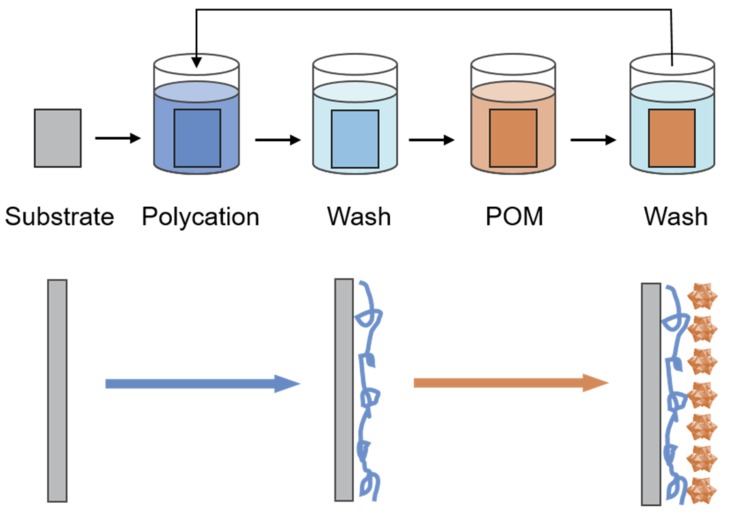
Schematic illustration of the fabrication procedure of POM-based multilayer films by layer-by-layer (LBL) self-assembly. Reproduced with permission from [66]. Copyright (2012) The Royal Society of Chemistry.

**Figure 4 molecules-24-03425-f004:**
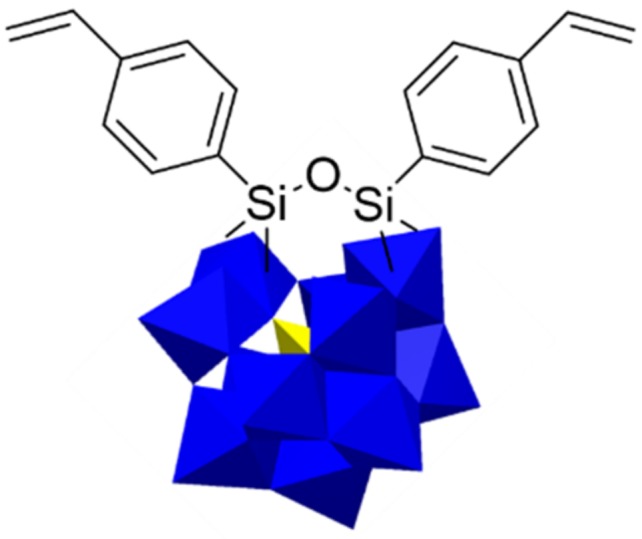
Structure of distyryl modified [SiW_11_O_39_]^4-^ via silanization. Reproduced with permission from [58]. Copyright (2019) Elsevier.

**Figure 5 molecules-24-03425-f005:**
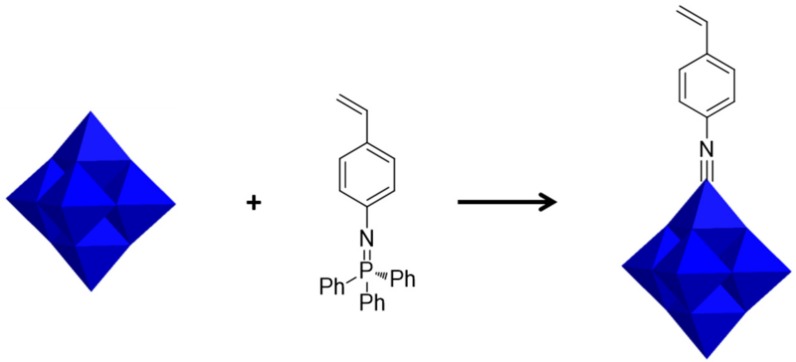
Preparation of POM-based monomer by imidization. Reprinted with permission from [58]. Copyright (2019) Elsevier.

**Figure 6 molecules-24-03425-f006:**
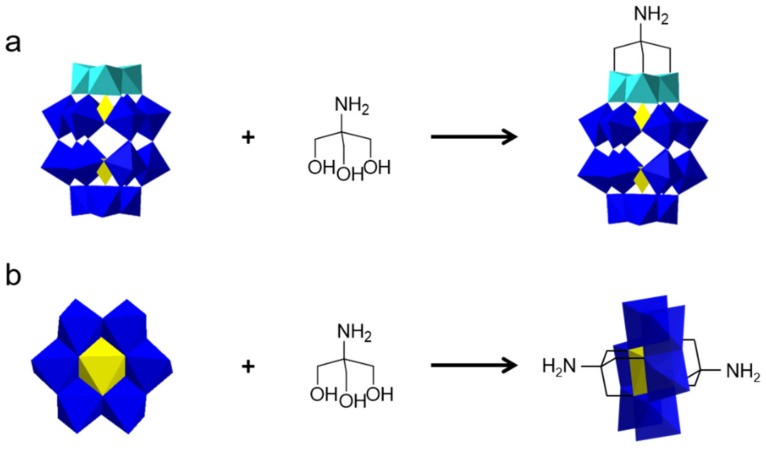
Preparation of POM-based monomer by esterification. (**a**) Dawson-type POMs; (**b**) Anderson-type POMs. Reproduced with permission from [58]. Copyright (2019) Elsevier.

**Figure 7 molecules-24-03425-f007:**
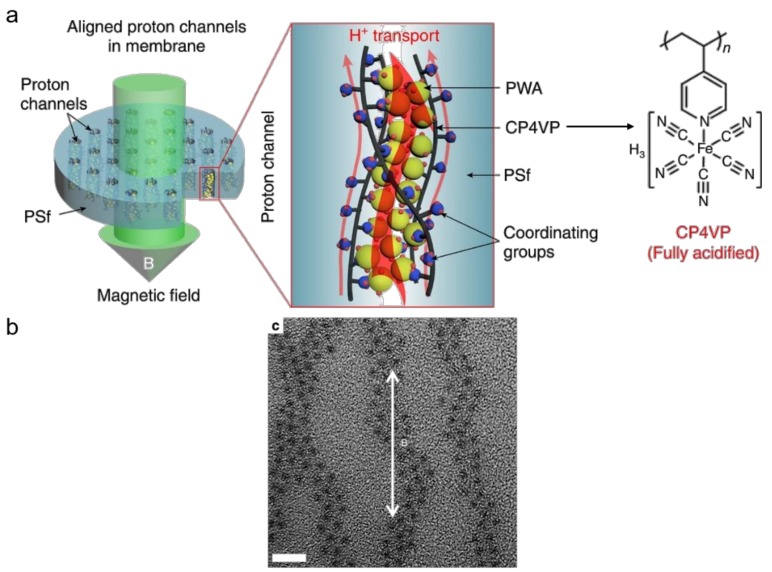
(**a**) Conceptual diagrams of magnetically aligned composite membrane and proton transport in the aligned channels in the membrane; (**b**) transmission electron microcsope (TEM) image for magnetically aligned composite membrane (5 nm scale bar), where the orientated chains of phosphotungstic acid (PWA) particles were clearly seen. Reproduced with permission from [62]. Copyright (2019) Creative Commons Attribution 4.0 International License.

**Figure 8 molecules-24-03425-f008:**
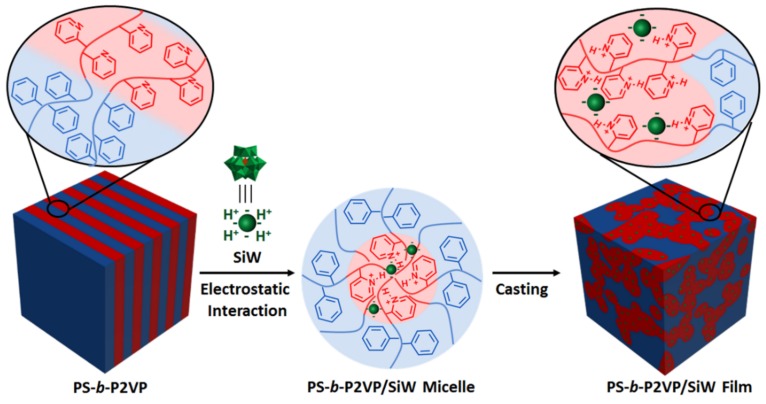
The SiW-induced phase transition of PS-b-P2VP from a lamellar to a bicontinuous structure upon casting the PS-b-P2VP/SiW micelle solution. Reproduced with permission from [46]. Copyright (2017) John Wiley and Sons.

**Figure 9 molecules-24-03425-f009:**
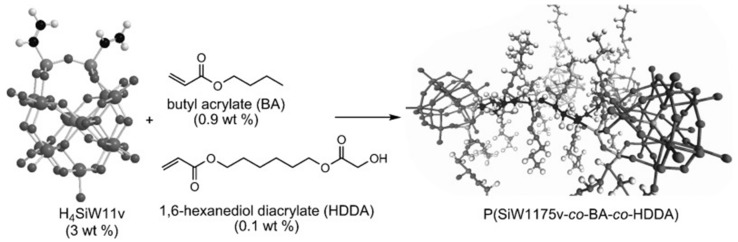
Synthesis of the copolymer (1,6-hexanediol diacrylate (HDDA) units in the copolymer are omitted for clarity). Reprinted with permission from [76]. Copyright (2009) John Wiley and Sons.

**Figure 10 molecules-24-03425-f010:**
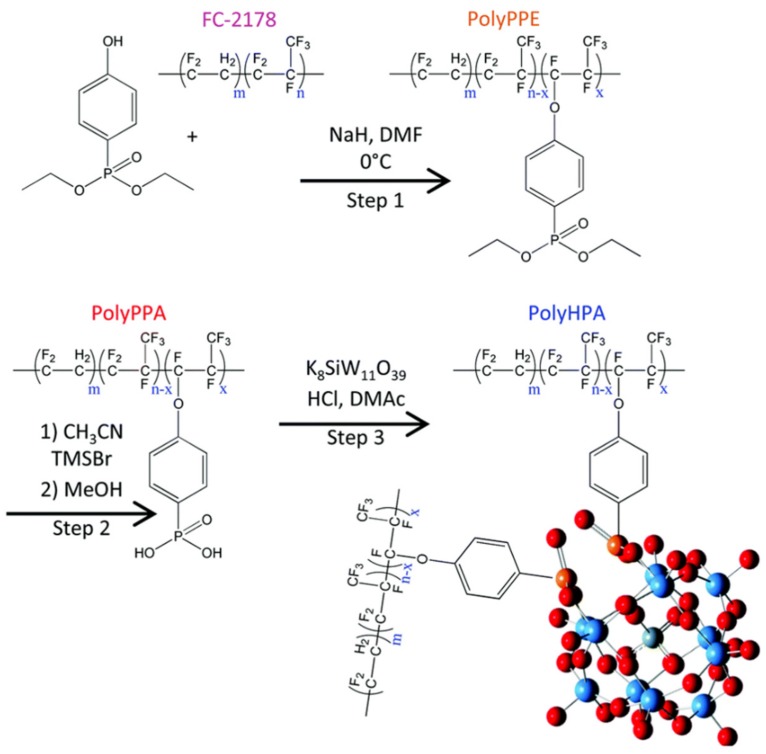
Full synthetic reaction scheme for the synthesis of PolyHPA. Reprinted with permission from [123]. Copyright (2018) The Royal Society of Chemistry.

**Table 1 molecules-24-03425-t001:** Summary of the proton conductivity of heteropolyacids (HPAs) with different hydration levels prepared in pellet or single crystal.

Compound	Conductivity (S cm^−1^) at Room Temperature	Reference
H_3_PW_12_O_40_·29H_2_O	0.17 (pellet)	[11]
H_3_PW_12_O_40_·28H_2_O	0.08 (single crystal)	[43]
H_3_PW_12_O_40_·21H_2_O	5 × 10^−3^ (pellet)	[39]
	7 × 10^−3^ (single crystal)	[43]
	1.8 × 10^−3^ (single crystal)	[44]
^a^H_3_PW_12_O_40_·14H_2_O	n.d.	[40]
^b^H_3_PW_12_O_40_·6H_2_O	n.d.	[40]
H_3_PW_12_O_40_	1.4 × 10^−6^ (pellet)	[42]
H_3_PMo_12_O_40_·29H_2_O	0.18 (pellet)	[11]
H_3_PMo_12_O_40_·21H_2_O	0.03 (pellet)	[39]
H_4_SiW_12_O_40_·28H_2_O	0.027 (single crystal)	[43]
^c^H_4_SiW_12_O_40_·6H_2_O	n.d.	[40]

^a, b, c.^ The exact values of the conductivity in these three pelletized compounds were not given directly, but shown in a figure of typical temperature dependent AC conductivity [40], in which the conductivity of them were in the range from 6 × 10^−5^ to 2 × 10^−4^ S cm^−l^ and in the order of a > c > b. n.d. = not determined.

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
