# Peer review of "Polyoxometalate–Polymer Hybrid Materials as Proton Exchange Membranes for Fuel Cell Applications"

_molecules, 2019, doi:10.3390/molecules24193425_

Round 1

Reviewer 1 Report

This paper reviews the current state of POM/polymer hybrids, and I suggests some aspects from a POM chemist.

1) In POM chemistry, typical heteroatoms are silicone or phosphorous, and we do not call tungsten or molybdenum as a heteroatom.  Those metals in POM are central role in POM, is is not hetero, but the difinition is of course somewhat ambiguous.

2)  There are excellent Pseudo liquid phase reviews in classic but it is missing.

3)Page 5, POMs do not decompose under pressure.

4) Some english adjective may be strange, such as detailedly, processability, and so on.

5) Page 7, the nucleophilicity of surface oxygen on POMs are generally considered to be really weak, especially in Keggin.  It is almost none.  It is a super strong acid, and just like SO42-, it is an anion part of strong sulfulic acid, it has a very weak coordination ability, or no coordination ability, and no nucleophilicity, in general inoganic chemistry common sense.

Author Response

Point 1: In POM chemistry, typical heteroatoms are silicon or phosphorous, and we do not call tungsten or molybdenum as a heteroatom. Those metals in POM are central role in POM, is not hetero, but the definition is of course somewhat ambiguous. 

 Response 1: We are sorry for making this mistake and thanks for the correction. We have corrected the statement by replacing "POMs are distinguished by the central heteroatoms M..." with "POMs, formed by the connection of {MOx} polyhedras (M = addenda atom, usually..." in the revised manuscript.

Point 2: There are excellent Pseudo liquid phase reviews in classic but it is missing.

Response 2: We are sorry for missing the important review about pseudoliquid phase. We have added a description of pseudoliquid phase with citing a classic review (Ref. [25]) in the revised manuscript.    

Point 3: Page 5, POMs do not decompose under pressure.

Response 3: We are sorry for the inaccurate expression and thanks for the correction. We have corrected the statement by replacing "The studies mentioned above were conducted on pressed powders of HPAs, which might decompose under pressure..." with "The studies mentioned above were conducted on pressed powders of HPAs, which exhibited high..." in the revised manuscript.

Point 4: Some English adjective may be strange, such as detailedly, processability, and so on.

Response 4: We are sorry for the improper expressions and thanks for the comments. We have changed these words in a more proper way in the revised manuscript.

Point 5: Page 7, the nucleophilicity of surface oxygen on POMs are generally considered to be really weak, especially in Keggin. It is almost none. It is a super strong acid, and just like SO42-, it is an anion part of strong sulfuric acid, it has a very weak coordination ability, or no coordination ability, and no nucleophilicity, in general inoganic chemistry common sense.

Response 5: We are sorry for the inaccurate expressions and thanks for your correction. We have removed the expressions of the nucleophilicity of surface oxygens on POMs in the revised manuscript.

Reviewer 2 Report

This comprehensive review submitted by Zhai and Li comprises the later advances on polyoxometalate-polymer hybrid materials for proton conducting membranes. After a brief introduction devoted to different mechanisms of protonation in heteropolyacids, the manuscript comments on the synthetic approaches that have been applied to date to prepare such hybrids and their performance as proton conducting devices. Overall the work is scientifically sound, clear and well organized. Therefore, I highly recommend its publication in Molecules. However, I would suggest a couple of minor modifications:

1) The authors state in line 78 that "POMs are distinguished by the central heteroatoms M..." This is not accurate. Heteropolyacids of general formula [XM12O40]n- are formed by the condensation of MO6 octahedra (M = addenda atom; usually W, Mo, V) surrounding a central heteroatom X (X = P, Si...). Please correct.

2) Line 97: Do the authors mean "protonation of shell/surface O atoms" when introducing "the unhydrated acidic proton"? Please explain.

3) Line 123: I do not understand the nature of the third class of protons. Please explain.

4) Line 131: Do the authors mean "protonation of shell/surface O atoms" when stating "free protons"? Please explain.

Author Response

Point 1: The authors state in line 78 that "POMs are distinguished by the central heteroatoms M..." This is not accurate. Heteropolyacids of general formula [XM12O40]n- are formed by the condensation of MO6 octahedra (M = addenda atom; usually W, Mo, V) surrounding a central heteroatom X (X = P, Si...). Please correct. 

 Response 1: We are sorry for making this mistake and thanks for the correction. We have corrected the statement by replacing "POMs are distinguished by the central heteroatoms M..." with "POMs, formed by the connection of {MOx} polyhedras (M = addenda atom, usually..." in the revised manuscript.

Point 2: Line 97: Do the authors mean "protonation of shell/surface O atoms" when introducing "the unhydrated acidic proton"? Please explain.

Response 2: We are sorry for the unclear expression, and thanks for the comment. "The unhydrated acidic proton" is the proton located on the surface oxygen atoms in the HPA anion. The precise position of this proton (the protonation site) has been controversial until the work of Janik and Deng et al. as we described in Line 99 and 100, "In anhydrous state, the position of the acidic protons of PWA were determined to be on both Oc and Od of the Keggin unit by Janik and Deng et al. theoretically and experimentally". (The number of Line corresponds to the revised manuscript)  

Point 3: Line 123: I do not understand the nature of the third class of protons. Please explain.

Response 3: We are sorry for the unclear expression. We supposed that you were confused about the nature of the third type of water in hydrated HPAs in Line 127. The third type of water is produced by protons and oxygen during decomposition of heteropolyacid and the statement was also explained in Line 134–136 with "In the range of 410–440 °C, a half or one water molecule, formed by acidic protons and oxygens from the host lattice lost, which resulted in the formation of denuded Keggin anion." We have slightly modified the statement of the three types of water in hydrated state by replacing "hydration water, protonized water composed of the crystalline water and protons..." with "hydration water (the crystalline water combined without protons), protonized water (the crystalline water combined with protons)...". (The number of Line corresponds to the revised manuscript)   

Point 4: Line 131: Do the authors mean "protonation of shell/surface O atoms" when stating "free protons"? Please explain.

Response 4: We are sorry for the unclear expression. The statement "free protons" in Line 134 is followed the description of the cited reference (Ref. [31]), which is similar to "the unhydrated acidic proton" because in anhydrous phase of PWA, there is no more crystalline water present. (The numbers of Line and Ref. correspond to the revised manuscript)